# Real Check RIO: A Real-World Analysis of Nivolumab in First Line Metastatic Melanoma Assessing Efficacy, Safety and Predictive Factors

**DOI:** 10.3390/cancers15041265

**Published:** 2023-02-16

**Authors:** Vlad-Adrian Afrăsânie, Teodora Alexa-Stratulat, Bogdan Gafton, Eliza-Maria Froicu, Daniel Sur, Cristian Virgil Lungulescu, Natalia Gherasim-Morogai, Irina Afrăsânie, Lucian Miron, Mihai-Vasile Marinca

**Affiliations:** 1Department of Medical Oncology, Regional Institute of Oncology, 700483 Iasi, Romania; 2Department of Oncology, Faculty of Medicine, “Grigore T. Popa” University of Medicine and Pharmacy, 700115 Iasi, Romania; 3Department of Medical Oncology, “Iuliu Haţieganu” University of Medicine and Pharmacy, 400015 Cluj-Napoca, Romania; 411th Department of Medical Oncology, “Iuliu Hatieganu” University of Medicine and Pharmacy, 400347 Cluj-Napoca, Romania; 5Department of Oncology, University of Medicine and Pharmacy of Craiova, 200349 Craiova, Romania; 6Department of Oncology, County Hospital of Emergency “Sf.Ioan cel Nou”, Bd. 1 decembrie 1918 nr. 21, 720224 Suceava, Romania; 7Department of Cardiology, Emergency Clinical Hospital “Sf. Spiridon”, Bd. Independenței nr. 1, 700111 Iasi, Romania

**Keywords:** metastatic melanoma, anti-PD-1 immune checkpoint inhibitor therapy, real-world experience, predictive biomarkers

## Abstract

**Simple Summary:**

With the recent advances of immunotherapy in the modern oncology era, the landscape of malignant melanoma has changed. However, it is not clear from the data across the various trials how to design the best therapeutic strategy. Hence, we decided to study the role of potential predictive biomarkers in a real-world, unselected population from Romania and to analyze the results of the single agent Nivolumab as first-line treatment in metastatic melanoma patients. Based on our findings, we believe that further studies could be designed to better map the immune landscape in metastatic melanoma in order to provide further perspectives on treatment options and choice.

**Abstract:**

We performed a retrospective study on 51 metastatic melanoma patients treated with Nivolumab in first line, at the Regional Institute of Oncology (RIO) Iasi, Romania between April 2017 and December 2019. We studied the efficacy and safety of anti-PD-1 immune checkpoint inhibitor therapy on a treatment-naive population. After a median follow-up of 36 months, the median progression free survival (PFS) was 26 months (95% CI, 15–36) and the median overall survival (OS) was 31 months (95% CI, 20.1–41.8). At 12 months after the initiation of immunotherapy, the percentage of patients alive was 70%, and at 24 months 62.5%. The most common adverse events observed were dermatological (23.5%) and grade ≥3 was identified in 4 (6.8%) patients. Multivariate analysis indicated that the presence of liver metastases (HR 4.42; 95% CI: 1.88–10.4, *p* = 0.001) and a neutrophils/lymphocytes ratio (NLR) were associated with poor survival (HR 3.21; 95% CI: 1.04–9.87, *p* = 0.04). Although retrospective data on a small group of patients were analyzed, we can conclude that our results in RIO are similar to those described in clinical trials and other real-world studies. Our study highlights the potential usefulness of liver metastases and NLR as novel predictive factors in clinical decision-making.

## 1. Introduction

With an estimated 325,000 new cases in 2020, malignant melanoma of the skin accounts for 1.7% of the global cancer burden [1]. While being only the third most common cutaneous malignancy, it is associated with the vast majority of skin cancer-related deaths. However, the incidence and mortality rates differ widely across the globe, depending on geography, ethnicity, socioeconomic context and other factors [2]. For example, data from population-based cancer registries in North America, Europe, Australia, and New Zealand suggest the incidence is higher in people up to age 50, who have photo-types I and II, numbers of nevi and prolonged exposure to ultraviolet rays mostly in their childhood [3]. An analysis of data from cancer registries found the lowest incidence rates in most regions of Africa and Asia, despite the high number of deaths per confirmed case [4].

Melanoma is an aggressive malignancy that tends to metastasize rapidly. Patients diagnosed with American Joint Committee on Cancer (AJCC) stage IV melanomas used to have a very poor prognosis, with a median overall survival (OS) of less than 1 year and a 5-year survival rate of less than 10% [5]. However, during the last decade, the game-changing class of checkpoint inhibitors have expanded melanoma survival to unprecedented rates. To evade immune surveillance, melanoma cells express co-inhibitory molecules within the tumor microenvironment, thus blocking an effective tumor clearance [6]. Inhibitory agents have been designed to target and block immune checkpoints. The cytotoxic T-lymphocyte–associated protein-4 (CTLA-4) blocking antibody ipilimumab and the programmed death 1 (PD-1) inhibitors (e.g., nivolumab or pembrolizumab) have significantly changed the treatment landscape and are currently an integral part of melanoma patient management, being used in both the metastatic and adjuvant setting with a confirmed improvement in OS rates [7]. For example, the 6.5-year OS data in the pivotal CheckMate 067 study showed improved clinical outcomes; furthermore, 77% of the patients treated with nivolumab plus ipilimumab and 69% of those treated with nivolumab monotherapy had not received subsequent treatment in prolonged follow-ups [8].

While efficacy and safety results from randomized clinical trials are crucial to regulatory approval, real-world data can provide valuable information about a drug in routine clinical practice. As individuals enrolled in trials are often quite different from real-world patients, clinical data, post-marketing assessments and updated data regarding safety and efficacy are paramount, especially for drugs that have only recently been approved. Real-world data is especially useful for identifying a drug’s true benefit for elderly patients that have significant comorbidities, a poor performance status, symptomatic brain metastases or any other poor-prognosis feature that precludes them from entering a clinical trial [9,10].

In Romania, immunotherapy for metastatic melanoma has been reimbursed since April 2017. Despite being used for more than five years, there are currently no regional data regarding checkpoint inhibitor efficacy and safety. The limited access to clinical trials is still a barrier which patients with cancer from Romania need to face; as a result, clinical research fails to become the backbone of treatment customization and the lack of evidence regarding adverse effects and consequences might affect therapeutic decision-making process [1,11,12].

We performed a retrospective analysis of all patients who received nivolumab in the first-line setting in a Romanian tertiary cancer center focusing on progression free survival (PFS), OS and toxicity. A secondary endpoint of our analysis was to identify potential predictive factors. Our study is one of the few carried out in Eastern Europe, to our knowledge, and of the first in Romania [13,14,15]. Thus, it can bring data about medical practice and outcomes in this geographic region for melanoma patients, which is different from Western Europe because of socio-economic conditions, and because of different biological and genetic traits [11].

## 2. Materials and Methods

We performed a retrospective analysis of all stage IV melanoma patients treated in the Regional Institute of Oncology (RIO) Iasi from April 2017 to December 2019. The follow-up period was defined in our study as the period between the diagnosis and death or loss of evidence or database closure (December 2021). The median follow-up period in our study was 23.3 months. Twenty-seven patients (52.9%) have died during the actual treatment time (April 2017–December 2021) and the follow-up period (January–December 2022). RIO Iasi is a 330-bed reference center for cancer patients in North-East Romania (roughly 5 million inhabitants) and one of the three cancer institutes in the country. In order to be included in the analysis, patients had to simultaneously meet the following inclusion criteria: age over 18 years, histopathology report of cutaneous malignant melanoma, stage IV disease certified by imaging or pathology results. Patients with mucosal melanoma or patients enrolled in clinical trials were excluded. Additional exclusion criteria were age under 18 years, absence of histopathology report, stage I-III of disease, patients who received any other treatment than nivolumab in the first-line setting. Multiple parameters were evaluated for this database: socio-demographical (age, sex), clinical (ECOG, primary tumor location, type of metastases, adverse events), biological (blood cell count values and derived indexes, glycemia and LDH), molecular (BRAF mutation status) and parameters specific for oncology (PFS and OS). The majority of parameters were collected before the initiation of the treatment.

OS was defined as the length of time between the date of nivolumab initiation and the time of death of any cause, and PFS was defined as the time from the of nivolumab initiation to the date of disease progression or death, whichever occurred first. Adverse events were recorded and graded according to The Common Terminology Criteria for Adverse Events (CTCAE) developed by the US National Cancer Institute (NCI) which is used to grade the severity of immune-related adverse events (irAEs) in both clinical trials and practice. IrAEs are classified into five grades, with 1 being the mildest and 5 indicating a high risk of death [16].

The objective of this study was to evaluate the efficacy and safety of nivolumab in first line treatment for patients with metastatic malignant melanoma. In addition, we tried to evaluate some parameters as predictive factors mainly derived from the red cell blood count. These were introduced in the database before the initiation of first cycle of immunotherapy in our hospital. The study was conducted in accordance with the Declaration of Helsinki, and approved by the Ethics Committee of the RIO Iasi. Due to its retrospective nature, rigorous anonymization and reporting of aggregate data, no specific patient consent was required.

All statistical analyses were performed using the SPSS version 25.0 software (IBM Corporation, Armonk, NY, USA). Patients not experiencing the relevant event or lost to follow-up were censored at the time of database lock. Both OS and PFS estimates were obtained using the Kaplan–-Meier method. Categorical variables were compared using the chi-square test. To identify potential predictive factors, we performed a univariate analysis using the log-rank test, and Cox regression was used to develop the multivariate model which included the resulting relevant candidates. Hazard ratios (HR) and 95% confidence intervals (95% CI) were computed, and the default threshold for statistical significance was set at 0.05 for all analyses.

## 3. Results

Between October 2017 and December 2021, 154 consecutive patients with de novo metastatic cutaneous melanoma were diagnosed in RIO Iasi. Of these, 76 were not treated with nivolumab, and 27 received it in the second or third lines of therapy. The remaining 51 patients were eligible for the analysis (Figure 1).

### 3.1. Patient and Tumor Characteristics

The clinical characteristics of the study population are summarized in Table 1. Thirty-two patients (62.7%) were male and 19 (37.3%) were female. The median age was 61.5 years, and most patients (*n* = 35, 48.6%) had an ECOG PS of 1, and only 1 (2%) patient had an ECOG of 3. For the studied population, the median PFS was 26 months (95% CI, 15–36) and the median OS was 31 months (95% CI, 20.1–41.8) (Figure 2 and Figure 3). At 12 months after the initiation of immunotherapy, the percentage of patients alive was 70%, and at 24 months 62.5%.

### 3.2. Survival Outcomes

Median PFS of the patient population reached 26 months (95% CI, 15–36), with 50% of patients not yet progressing after 2 years (Figure 2). Median OS was 31 months (95% CI, 20.1–48.8); the landmark 12-months and 24-months survival were reached by 70% and 62.5% of patients, respectively (Figure 3).

### 3.3. Immune Related Adverse Events

IrAEs were recorded in 40 of 51 (83%) melanoma patients treated with immune checkpoint inhibitors (ICIs) in our hospital (Table 2). The most commonly observed irAEs were dermatological (12 patients, 23.5%) and endocrinological (8 patients, 15.6%); pulmonary and gastrointestinal events were experienced by 2 patients each, while grade ≥3 irAEs occurred in only 4 cases (6.8%).

### 3.4. Predictive Factors

Multivariate analysis indicated that the presence of liver metastases (HR 4.42; 95% CI 1.88–10.4, *p* = 0.001) and a high neutrophil-lymphocyte ratio (NLR) are independent prognostic factors for a decreased OS (HR 3.21; 95% CI 1.04–9.87, *p* = 0.04) (Table 3). Patients without liver metastases had a median OS of 40 months compared with 11 months in patients with liver metastases (Figure 4).

We selected 3 as the cut-off value for NLR in this study because it was very close to our mean and median values, which were 3.14 and 2.45 respectively. Another justification was that it has been cited in other research. A NLR equal or above 3 was associated with poor prognosis; subjects with NLR equal or above 3 had a lower OS compared with those with a NLR below 3 (19 versus 41 months). (Table 3, Figure 5).

In our study population, patients with low NLR had longer OS, as did those without liver metastases, and differences did not vary between groups with and without liver metastases (the NLR < 3.0 group—which was also the largest—had higher OS regardless of liver involvement). However, Levene’s test of variances yielded a *p*-value of 0.017, which implies interaction is present (probably in the group of patients with liver metastases, but this would have to be verified on a larger sample).

## 4. Discussion

In the management of advanced melanoma, both BRAF/MEK targeted agents and immunotherapy are approved. The first turning point in the therapeutical approach for melanoma was the discovery of activating BRAF mutation. BRAF testing is recommended at the time of advanced melanoma diagnosis and is present in approximately 50% of cases, information also confirmed by data from real-world studies [11,17,18,19]. Despite the high frequency of BRAF mutated cases reported in the literature, our cohort describes the presence of only 9.8% (*n* = 5) patients and in all BRAF mutant cases V600E mutations were identified. This might be explained by the limitations of our study, namely the retrospective nature of the study, the small number of subjects and the patient selection bias.

The discovery of checkpoint inhibition and the 2011 FDA approval of ipilimumab in metastatic malignant melanoma, a monoclonal antibody that blocks CTLA-4, marked the beginning of a new era in oncology. Subsequently, nivolumab—a PD-1 inhibitor for the treatment of malignant melanoma—was also approved by the FDA in 2014. These two molecules have revolutionized the treatment of metastatic malignant melanoma and have allowed patients to achieve previously unimaginable survival rates [20,21].

Immunotherapy has yielded remarkable results for patients with metastatic malignant melanoma in Phase III trials. However, it is not known whether these results can be translated to an unselected population in Romania. In the Phase III registration trials CheckMate 066 and CheckMate 067 in which patients with metastatic malignant melanoma were treated with nivolumab in the first line of treatment a median OS of 37.3 months after 5 years of follow-up was achieved, and 34.4 months after 6.5 years of follow-up in the second trial. OS at 12 months was 71% in the first trial, and 75% in the second trial. OS at 24 months was 58% in CheckMate 066 and 59% in CheckMate 067 at 24 months. At 36 months in CheckMate 066 it was 51%, and 46% in CheckMate 067; at 5 years it was 39% and at 6.5 years 44% [8,22]. Studies including real-world patients from Poland, USA, France and England had an OS at 1 year of about 70–80% and 45–75% at 24 months [18,23,24,25]. In our study, the efficacy of nivolumab was comparable to data from clinical registration trials and data from other real-world studies; thus, the median OS was 31 months, and the OS at 12 months was 70% and at 24 months 62.5%.

Nivolumab is a treatment with a good safety profile, with specific adverse effects being immune-mediated. In the Checkmate 066 trial, immune-mediated adverse events had an incidence of 74.3% [20]. In the CheckMate 067 trial the most common immune-mediated adverse effects were rash (16%), hypothyroidism (12%) and diarrhea (5%). Of these, grade 3 or 4 immune-mediated adverse effects were diarrhea (4%), hepatitis (3%) and rash (2%) [21]. Xing et al. published the results of a meta-analysis evaluating the adverse effects of nivolumab and ipilimumab and nivolumab monotherapy in 7396 patients. The most common immune-mediated adverse effects for nivolumab were pruritus (12.1%), diarrhea (11.1%) and rash (11.06%). The most common grade 3 or higher adverse effects were pneumonitis (2.6%), pancreatitis (1.6%) and hyperglycemia (0.9%) [26].

Our research found that adverse events were in a proportion of 54.9%, most commonly dermatological (in 23.5% of patients) and endocrinological (in 15.6% of patients). Although irAEs were common, their advantage was that only a low proportion (6.8%) were grade 3 and 4. Compared to the literature, in our study population irAEs were more common than in the registration trials and in the meta-analysis, but with a severity of irAEs that is comparable. The greater disparity is attributable to a increased occurrence of dermatological adverse effects, which can be ascribed to the study’s limitations. Although the results with nivolumab in metastatic malignant melanoma are promising, and 44% of patients are still alive 6.5 years after treatment initiation, we do not have biomarkers to predict which patients will benefit or not from treatment [21]. Half of the patients will not benefit from nivolumab and for this reason they need to be identified in order to find other therapeutic strategies that could prolong their survival. Given the risk of nivolumab non-responders being deprived of other effective treatment and being exposed to toxicities, and also to add financial burden to the healthcare system, there is an unmet need for predictive biomarkers [27].

In the current study, therefore, we focused on trying to find possible predictive factors. The literature already describes the role of tumor mutational burden (TMB) as a genomic biomarker which predicts a better response to ICI and a favorable survival rate. Therefore, additional tools with clinical feasibility can be investigated [28]. In the multivariate analysis, liver metastases (LM) and NLR had predictive value. Thus, the presence of hepatic secondary lesions was independently associated with decreased OS (HR = 4.42, 95% CI, 1.88–10.4, *p* = 0.001).

Patients with liver metastases had a median OS of 11 months compared to 40 months for patients without liver metastases. It is not new that, regardless of tumor histology, the presence of LM translates as an indicator of poor prognosis. More than that, it was noticed that patients with LM had a higher number of distant metastatic sites. Many studies have concluded that patients with LM had a shorter OS and PFS [29,30]. Similar data have been reported in non-small-cell lung cancer (NSCLC) patients who received PD-1/PD-L1 monotherapy; those with LM had considerably shorter OS, about 10 months, than those without LM, with 20 months. A meta-analysis of 8 randomized clinical trials investigating the efficacy of PD-1 and PDL-1 inhibitor immunotherapy in various cancer types with LM found no statistically significant association between them. However, no randomized clinical trials which included patients with malignant melanoma were analyzed [31]. There is little information in the literature examining the effects of immunotherapy in patients with metastatic malignant melanoma and liver metastases. Tumeh et al. examined the relationship between LM and PFS in a malignant melanoma population and a worse outcome was described for the LM group (2.7 months vs. 18.5 months, HR = 1.85, 95% CI, 1.37–2.5, *p* < 0.0001). A worse objective response rate was also demonstrated for patients with LM than those without [30]. Ma et al. conducted a single center retrospective study of 327 patients with metastatic melanoma who received immunotherapy and showed that patients without LM had a better survival at 3 years compared with those with LM, 28.7% vs. 74.4%, HR = 2.1, 95% CI, 1–5.3.1. In their research the combination of nivolumab and ipilimumab was more effective in patients with LM [32]. One explanation of these findings may be because of the liver’s unique blood supply, which enables an immune tolerogenic microenvironment to be shaped. As primary tumors have been the main focus in previous studies, the characteristics of the local immune status in metastatic tumors are not clearly understood or described. Despite this, there are proofs that demonstrate the connection between a low infiltration of CD8+ T cell at the invasive margin and the presence of LM as a low prognostic response to anti-PD-1 therapy. Furthermore, the LM group was also found to have lower CD8+ T-cell infiltration in extrahepatic metastatic lesions compared with patients without LM. These results suggest LM may inhibit antitumor immunity by promoting a systemic immunosuppressive effect. Likewise, more associated factors such as female gender, high levels of LDH, and prior administration of ipilimumab were linked to a supplementary risk of inferior clinical response [30].

NLR is defined as the ratio of neutrophils to lymphocytes and is a prognostic factor in several neoplasms, but the mechanism is still unclear. Previous studies have shown that inflammation in cancer has been recognized as one of the markers with a critical role in modulating the tumor microenvironment [27,28]. Increased levels of systemic inflammation have been correlated with reduced survival and poor response to treatment in several solid tumors [33,34]. Inflammation may play an important role in tumor genesis and progression by promoting tumor cell proliferation and survival, angiogenesis, and tumor metastasis, but also as a factor influencing tumor response to systemic therapies [35]. Systemic inflammation in patients with malignancies is thought to shape the profile of cytokines produced both by the tumor and as a component of the host response to tumor aggression [36]. Lymphocytes, thought to play a key role in immune surveillance, have a role in the maturation of tumor suppression [37]. The promotion of tumor growth by the immune cells from the tumor microenvironment may also contribute in the inflammatory response. Neutrophils recruited to the tumor stroma exert pro-tumorigenic effects by inhibiting apoptosis, promoting angiogenesis and stimulating metastasis formation [38] and by creating an inflammatory microenvironment favorable for malignant growth. CD4+ T helper cells and CD8+ cytotoxic T cells are known to play an important role in immunosurveillance. However, there is considerable controversy regarding the mechanisms by which NLR might impact the prognosis of cancer patients. Several explanations could be provided. Preclinical studies have shown that neutrophils not only promote cancer cell metastasis, invasion and proliferation, but also help the cells to evade immune surveillance, thereby promoting tumor activity [39]. Lymphocytes possess anti-tumor activities that may induce cell death, inhibit tumor cell migration and growth and are the main inflammatory factors that inhibit cancer cell progression [40]. This could explain why an increased NLR is associated with a poor prognosis in cancer patients. There are no prospective studies that validate a universal cut-off for the NLR value.

At this time, the optimal value for the NLR cut-off is unknown, with investigators reporting values between 2 and 5 [40,41]. An NLR above or equal to 3 was highly associated with poor OS (HR = 3.21, 95% CI, 1.04–9.87, *p* = 0.04).

The median OS of patients with a NLR above or equal to 3 was 19 months compared to 40 months for the patients with a NLR below 3. Taking into account the median and mean for NLR in our study and the results reported from other studies, in this analysis we chose to use the value of 3 as a cut-off [42,43]. There is a need for prospective randomized trials to validate the cut-off value for NLR in patients with metastatic malignant melanoma receiving systemic treatment.

In clinical trials an increased NLR has been associated with reduced survival in patients with gastrointestinal cancers, pancreatic cancer, breast cancer, bladder cancer and other cancers. A recent meta-analysis of 40,599 patients with solid tumors showed that an NLR greater than 4 was associated with a substantially increased risk for all-cause mortality [44,45]. However, the cutoff value and prognostic effect of NLR and platelet/lymphocyte index remain controversial. It has been shown that the optimal cutoff value for prognostic indicators could be chosen from selected values from other studies [31,39]. Among a meta-analysis that included 13 studies, Zhang et al. examined the predictive role of NLR in 2208 patients with metastatic malignant melanoma. An increased NLR was associated with reduced OS in patients treated with anti-PD-1 (NLR: HR-2.42, 95% CI, 1.68–3.5). Although the predictive role of NLR in melanoma patients receiving immunotherapy has begun to be recognized and investigated in recent years, there are open questions regarding the standardized value of NLR. Zhang’s meta-analysis found that although most studies used a cut-off value of NLR = 5, more than one cut-off value could be employed. All studies showed that reduced OS was associated with a NLR above the chosen cut-off point [46].

There is a large body of preclinical and clinical data that supports the role of NLR as a predictive and prognostic marker in various neoplasms, with more recent studies focusing on malignant melanoma. This was also demonstrated in our study.

It is necessary to take into account the several potential limitations of the retrospective nature of the study, including the limited observational time, incomplete data, possible underreporting of adverse events not uniformly recorded for all subjects. Other aspects we need to consider are the small number of subjects and the selection bias which necessitated a careful interpretation of our findings.

The strength of our study includes the observation of treatment-naïve patients, outside of a clinical trial protocol. Our results support the favorable outcomes of ICI treatment for advanced melanoma patients in a real-life setting and demonstrate that there are more benefits than risks in a clinically diverse patient population, including less fit and even frail older adults, who may not have otherwise been eligible for the phase 3 studies.

Furthermore, multivariate analysis accounts for multiple known prognostic variables that may impact immunotherapy outcomes in metastatic melanoma. Our research has several limitations, mostly related to the retrospective design, relatively low number of patients and the single-institution setting. Further prospective studies including larger numbers of patients and biomarkers associated with ICI efficacy and safety are needed to validate our observations.

## 5. Conclusions

This study represented the first instance of this research being conducted in Romania, and is one of few similar studies done in Eastern Europe. Our research confirms the efficacy and safety of nivolumab in metastatic melanoma patients; the results found are comparable to those from clinical trials, or from other populations included in real-world studies. This study also explored findings from previous research on predictive factors for nivolumab, and demonstrated that the presence or absence of LM and NLR could be used to select patients who would most likely benefit from immunotherapy as first-line treatment.

## Figures and Tables

**Figure 1 cancers-15-01265-f001:**
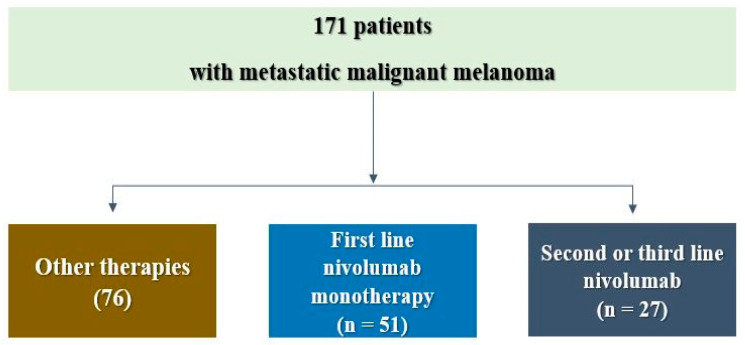
Flow diagram of the study population selection process.

**Figure 2 cancers-15-01265-f002:**
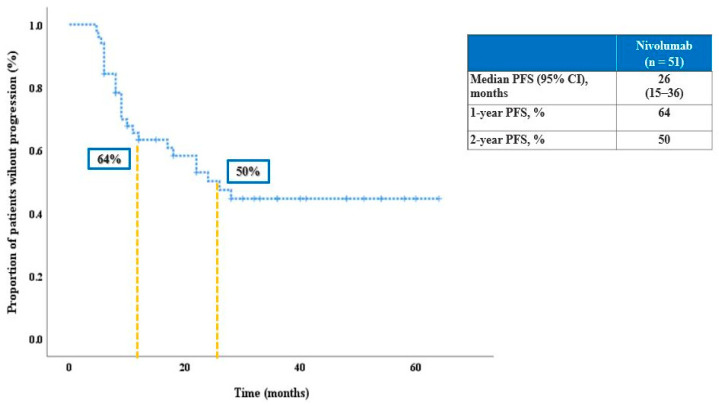
PFS of studied population.

**Figure 3 cancers-15-01265-f003:**
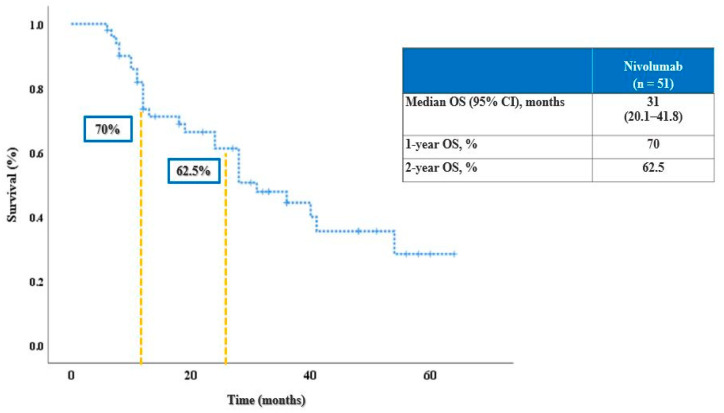
OS of studied population.

**Figure 4 cancers-15-01265-f004:**
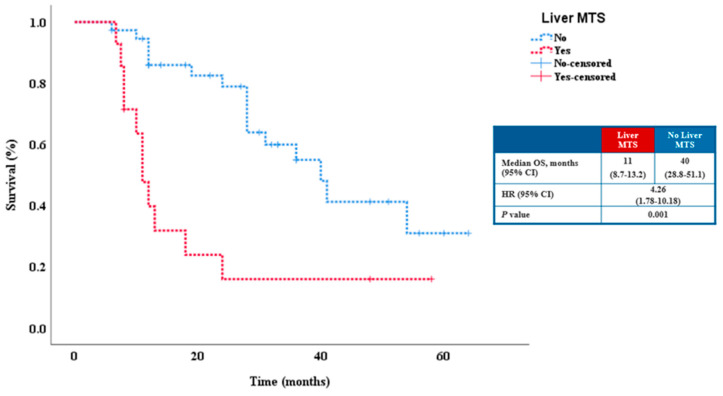
Relationship between OS and presence of liver metastases in the study population.

**Figure 5 cancers-15-01265-f005:**
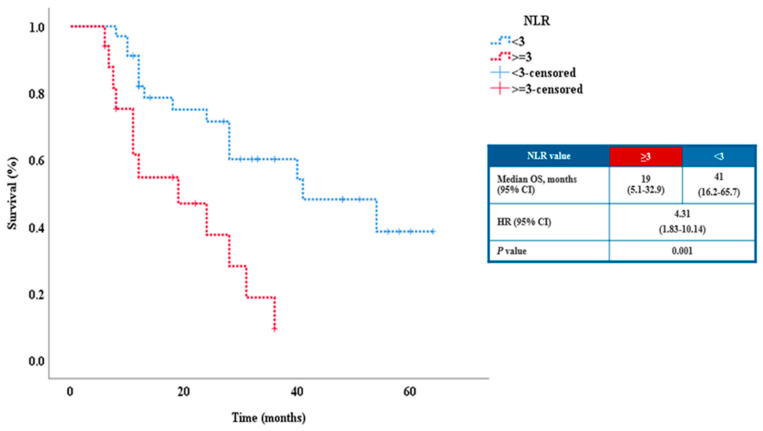
Relationship between OS and NLR value in the study population.

**Table 1 cancers-15-01265-t001:** Characteristics of the study population.

Characteristic		No. of Patients (*n* = 51)
Mean age (years)	61.5 (range 18–88)	
Gender, *n* (%)	Male	32 (62.7)
	Female	19 (33.3)
ECOG PS, *n* (%)	0	9 (17.6)
	1	35 (68.6)
	≥2	7 (13.8)
Primary tumor location, *n* (%)	Head	19 (37.3)
	Thorax	21 (41.2)
	Extremities	8 (15.7)
	Unknown	3 (5.9)
Metastatic sites *, *n* (%)	Lung	21 (42)
	Liver	14 (27.5)
	Skin	9 (17.6)
	Brain	8 (15.7)
	Bone	7 (13.7)
LDH level, *n* (%)	Normal (120–246 U/L)	37 (72.5)
	High (>246 U/L)	14 (27.5)
Pathology subtype, *n* (%)	Ulcerated	16 (31.4)
	Nodular	35 (68.6)
BRAF mutation status, *n* (%)	Wild-type	46 (90.2)
	Mutant	5 (9.8)

* Some patients had more than one metastatic site. ECOG PS, Eastern Oncology Cooperative Group performance status; LDH, lactate dehydrogenase; BRAF, serine/threonine-protein kinase B-Raf.

**Table 2 cancers-15-01265-t002:** Immune related adverse events (irAE).

Adverse Event	Any Grade (*n*, %)	Grade 3 and 4 (*n*, %)
Any	28 (54.9%)	4 (6.8%)
Dermatological	12 (23.5%)	2 (3.9%)
Gastrointestinal	2 (3.9%)	0 (0.0%)
Rheumatological	4 (7.8%)	0 (0.0%)
Endocrinological	8 (15.6%)	2 (3.9%)
Pulmonary	2 (3.9%)	0 (0.0%)
Renal	0 (0.0%)	0 (0.0%)

In the univariate analysis, the variables correlated with a decreased OS were the presence of liver metastases, the occurrence of irAEs of any grade, the neutrophil-lymphocyte ratio (NLR), and the systemic inflammatory index (SII).

**Table 3 cancers-15-01265-t003:** Univariate and multivariate analysis of factors influencing OS.

	Univariate Analysis			Multivariate Analysis		
	HR	95% CI	*p* Value	HR	95% CI	*p* Value
Age	1.01	0.98–1.04	0.44			
Gender	0.55	0.23–1.31	0.18			
ECOG PS	1.01	0.56–1.82	0.95			
Primary tumor location	1.16	0.81–1.66	0.41			
BRAF mutation	0.63	0.15–2.68	0.53			
Lung metastases	1.13	0.53–2.42	0.74			
Liver metastases	3.5	1.59–7.69	**0.002**	**4.42**	**1.88–10.4**	**0.001**
Cutaneous metastases	0.95	03.5–2.53	0.92			
Brain metastases	1.12	0.42–2.98	0.8			
Bone metastases	1.03	0.35–3.04	0.94			
irAEs	0.35	0.13–0.95	0.04	0.31	0.18–1.74	0.56
High LDH	1	0.99–1	0.71			
Leukocytosis	0.93	0.41–2.13	0.93			
Neutrophilia	1.35	0.61–2.97	0.44			
Lymphocytosis	0.77	0.18–3.32	0.73			
Monocytosis	1	0.99–1	0.16			
Thrombocytosis	2.32	0.67–8.08	0.18			
Hyperglycemia	1.5	0.67–3.34	0.31			
NLR	3.65	1.63–8.17	**0.002**	**3.21**	**1.04–9.87**	**0.04**
TLR	1.83	0.85–3.94	0.11			
TWR	0.89	0.4–1.96	0.78			
MWR	0.91	0.42–0.19	0.82			
MLR	0.57	0.26–1.21	0.14			
SII	2.33	1.06–5.13	0.03	1.25	0.41–3.78	0.69

ECOG PS, Eastern Cooperative Oncology Group performance status; BRAF, serine/threonine-protein kinase B-Raf; irAE, immune related adverse event; LDH, lactate dehydrogenase; NLR, neutrophils-lymphocytes ratio; TLR, thrombocytes-lymphocytes ratio; TWR, thrombocytes-white blood cell ratio; MWR, monocytes-white blood cell ratio; MLR, monocytes-lymphocytes ratio; SII, systemic inflammatory index.

## Data Availability

The data presented in this study are available on request from the corresponding author. The data are not publicly available due to privacy restrictions.

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
