# Peer review of "Real Check RIO: A Real-World Analysis of Nivolumab in First Line Metastatic Melanoma Assessing Efficacy, Safety and Predictive Factors"

_cancers, 2023, doi:10.3390/cancers15041265_

Round 1

Reviewer 1 Report

The manuscript by Afrăsânie et al., titled "REAL CHECK RIO: a Real-world Analysis of Nivolumab in First Line Metastatic Melanoma Assessing Efficacy, Safety, and Predictive Factors," is an intriguing study with important implications in the current clinical clinical setting. Some points are unclear to me and require clarification from the author. The following are some comments:

1. The author of M & M mentioned the status of BRAF mutation in their cohort study. But they didn't say whether all BRAF mutations were BRAFV600E. BRAFV600E mutation was found in more than 40% of melanomas.

2. Separating the results into separate sections will make the manuscript easier to read.

3. How many patients died during their research?

4. It is recommended that the abbreviation be explained the first time it is used. For example, the terms OS and PFS were used in the abstract but their full form was mentioned in M & M.

5. Specify immune-related adverse events (irAEs). How the author classifies/grades irAEs is not explained in the manuscript.

6. The manuscript lacks details on NLR grading.

Figure 5 has blurriness. Please change the figure.

8. Cite tables in the text in the same way that you mention them in the legend (i.e. if you mentioned it Table 2 then in the text it should be cited as Table 2 not in like Table II).

9. There are a lot of typos; spell check is required.

Reviewer 2 Report

The authors presented a manuscript entitled: “REAL CHECK RIO: a Real-world Analysis of Nivolumab in 2 First Line Metastatic Melanoma Assessing Efficacy, Safety and 3 Predictive Factors.” They performed a retrospective study on metastatic melanoma patients treated with the anti-PD1 immune checkpoint inhibitor Nivolumab in first line. They aimed to study the efficacy and safety of Nivolumab and to identify some predictive biomarker for patients’ responsiveness to the treatment. 

The manuscript is very interesting and well written and suitable for publication.

I just suggest paying attention to the abbreviations: they must write literally the words and the abbreviation the first time they mention it, otherwise it is not clear what they are referring to (es. Abstract line 32, Introduction line 90, etc.). Moreover, there’s some typos and repetition in the text (es. Line 16-128, line 307). 

Round 2

Reviewer 1 Report

The authors responded satisfactorily to the comments. The manuscript is ready for publication in the prestigious Cancers-MDPI journal.